# Mitigating Catastrophic Forgetting with Context-aware Continual Pretraining for LLMs

## Abstract

Retraining large language models (LLMs) from scratch to include novel, internal or domain-specific knowledge is prohibitively computationally expensive. Therefore, practitioners rely on continual pretraining to adapt existing pretrained models to new data. As the model's parameters are updated to assimilate new information, it can abruptly lose proficiency on previously learned domains, a phenomenon known as catastrophic forgetting. To address this issue, we propose Context-aware Continual Pretraining (CA-CPT), a simple technique that provides the model with sample-specific context before adapting its weights to new content in order to smoothen the training loss. Our empirical results demonstrate that CA-CPT has comparable or superior performance on new domain data while consistently mitigating the forgetting of both general knowledge and specialized instruction-following abilities. We show that our method is broadly applicable, is orthogonal to existing catastrophic forgetting mitigation strategies, and can serve as a building block for more robust continually learning language models.

## 1 Introduction

Large language models (LLMs) have established themselves as cornerstones of modern natural language processing, in part due to their scalability (Brown et al., 2020; Hoffmann et al., 2022; Kaplan et al., 2020). LLMs have billions of parameters and are pretrained on a corpora of trillions of tokens, an extensively compute-intensive process. The availability of high-performing open-weights models (Abdin et al., 2024; Grattafiori et al., 2024; Jiang et al., 2024; Liu et al., 2025; Riviere et al., 2024; Walsh et al., 2025; Yang et al., 2025) has democratized LLMs, enabling practitioners to build upon existing foundations.

The static nature of these foundation models presents a critical limitation. To maintain their relevance and utility, particularly for specialized applications in fields like finance, law, or medicine, LLMs must be continuously updated with new knowledge. Retraining the model from scratch on a combined corpus of old and new data is computationally and financially infeasible for all but a few organizations. This economic reality has given rise to a technique known as continual pretraining (CPT) (Jin et al., 2022). This can come at the cost of decreased performance on previously known domains, a phenomenon known as catastrophic forgetting (McCloskey & Cohen, 1989; Ratcliff, 1990; van de Ven et al., 2024). The model, in its effort to minimize the loss on the new data, aggressively updates its parameters in a way that disrupts the delicate configuration responsible for encoding prior knowledge. This challenge is a manifestation of the classic stability-plasticity dilemma (Grossberg, 1987; Mermillod et al., 2013). Simultaneously, a system must be stable, to preserve existing knowledge, and plastic, to learn new information. Navigating this trade-off is the central goal of continual learning research.

In this work, we present Context-aware Continual Pretraining (CA-CPT), a continual pretraining method designed to mitigate catastrophic forgetting based on the observation that the initial tokens of a sequence have a disproportionately high loss (Section 3.1), which is detrimental to the stability-plasticity tradeoff. Our method directly addresses this issue by strategically masking these high-loss initial tokens. It is a data-processing technique that can be seamlessly combined with other

approaches, such as replay-based methods, regularization techniques or architectural methods. Our main contributions are summarized as follows:

- We propose CA-CPT, a data-centric continual pretraining approach designed to mitigate catastrophic forgetting in LLMs following continual pretraining.

- We provide a theoretical analysis showing that masking initial tokens reduces gradient variance, thereby enhancing training stability and improving the stability–plasticity trade-off.

- Empirically, we demonstrate that CA-CPT not only reduces catastrophic forgetting but also enables efficient knowledge acquisition from new, domain-specific data.

- We show that CA-CPT is complementary and orthogonal to existing continual learning techniques, making it broadly applicable.

## 2 RELATED WORK

### 2.1 CONTINUAL LEARNING & PRETRAINING

The challenge of adapting large models to new data streams without erasing prior knowledge has spurred a rich and diverse field of research. Comprehensive surveys provide a structured taxonomy of continual learning approaches (Shi et al., 2024a; Wang et al., 2024; Wu et al., 2024b), which we broadly categorized into four categories: replay, regularization, architecture and training regime.

**Replay-Based Methods.** The most intuitive approach to preventing forgetting is to periodically rehearse previously learned information. Replay-based methods achieve this by storing a small subset of data from past tasks and interleaving these samples with new data. Simple replay has been shown to be an effective baseline when continually pretraining LLMs (Ibrahim et al., 2024). Replay can also be done generatively, where a model learns to produce synthetic data from past tasks (Shin et al., 2017).

**Regularization-Based Methods.** These techniques modify the learning objective by adding a penalty term to the loss function discouraging significant changes to model parameters. For example, Elastic Weight Consolidation (Kirkpatrick et al., 2017) selectively makes learning slower on weights deemed important for previous tasks. Such methods have been proved effective for continually learning language models (Rongali et al., 2021).

**Architectural Methods.** Some parameter-efficient techniques like Adapters (Houlsby et al., 2019) and Low-Rank Adaptation (LoRA) (Hu et al., 2021) freeze the main model and only train small new modules. Similarly, LLaMa Pro (Wu et al., 2024a) duplicates transformer blocks and trains the new blocks on new corpus, allowing new capacity for new knowledge.

**Training Regime Based Methods.** Gupta et al. (2023) study the importance of rewarming the learning rate when continually pretraining from a checkpoint with a decayed learning rate. Ibrahim et al. (2024) establish that a simple combination of learning rate rewarming followed again by decay has a great effect when paired with data replay.

Our work proposes a data-centric strategy, which is orthogonal to these approaches and tailored to the unique challenges of continual pretraining of LLMs.

### 2.2 DATA SELECTION

By curating the data stream, it is possible to create a more effective and stable learning signal. We present two levels of granularity for data selection: the sample level and the token level.

**Sample-Level Data Selection.** Sample-level data selection methods operate on entire documents or sequences (Albalak et al., 2024). One well-established strategy is curriculum learning (Bengio et al., 2009), where the model is first trained on easier examples before being exposed to more

complex data. The intuition is that this gradual increase in difficulty provides a more stable learning trajectory. Other methods focus on the ordering of data to maximize contextual learning. For instance, In-context Pretraining reorders documents so that semantically related documents appear consecutively within the model's context window, encouraging it to learn across documents (Shi et al., 2024b). Similarly, LinkBERT constructs training sequences by connecting documents via hyperlinks, treating the corpus as a graph (Yasunaga et al., 2022).

**Token-Level Data Selection.** Token-level strategies operate at a finer granularity, making decisions about which individual tokens to include in the learning objective. The most foundational form of token-level selection is the Masked Language Modeling (MLM) objective introduced with BERT (Devlin et al., 2019). In MLM, a random subset of tokens is replaced with a special masked token and the model is trained to predict the original tokens. Recent work like Rho-1 introduced Selective Language Modeling (SLM) (Lin et al., 2025). SLM selectively trains on tokens that are deemed most "useful" by calculating an "excess loss" for each token relative to a smaller, high-quality reference model. More targeted approaches have been proposed for domain adaptation. Gu et al. (2020); Lad et al. (2022) selectively mask important tokens to learn domain-specific patterns during a second pretraining phase. Other methods, instead of specifically learning the most important tokens, will opt to ignore the least important ones. Hou et al. (2022) aim to reduce training time by dropping unimportant tokens, but can fall short in handling semantic knowledge tasks (Zhong et al., 2023).

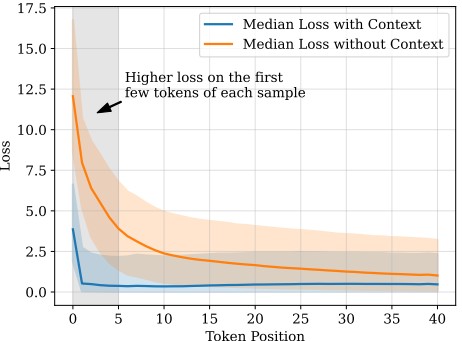 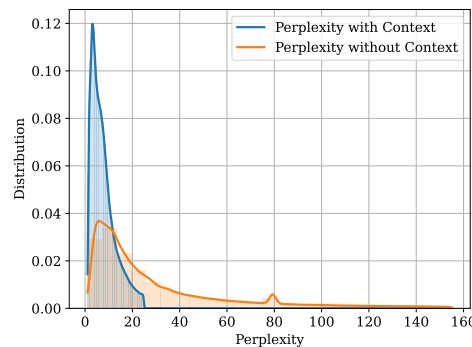

(a) Median Loss and Interquartile Range by Token Position Across Various Domain Adaptation Datasets.

(b) Perplexity Distribution Across Various Domain Adaptation Datasets.

Figure 1: The Impact of Context on the Loss and Perplexity Distribution.

## 3 METHODOLOGY

In this study, we define the "context" of a sample as its beginning, which consists of a few tokens, can vary in length and that will be masked for the training loss. We define the "content" of a sample as the rest of the sample that we use to calculate the loss and train our model.

### 3.1 INITIAL OBSERVATION

Our approach is motivated by a key empirical observation: when training large language models, the per-token loss is consistently and significantly higher during the initial tokens of a sequence. We believe that in the context of CPT, where the model has already acquired extensive general linguistic knowledge, allowing the training update to be heavily influenced by this initial high loss is inefficient. This spike in loss at the beginning of a sequence is often an artifact of the model's limited context at that point, rather than a true indicator of its inability to generate accurate content. This phenomenon is clearly illustrated in Figure 1.

To investigate this phenomenon further, we extend this observation to the overall perplexity of a sample. By applying a masking strategy to the first few tokens of a sequence, we observed a drastic

improvement in the sample's overall perplexity. This finding suggests that mitigating the influence of these high-loss tokens can lead to more stable and efficient training. Based on these observations, we hypothesize that CA-CPT, which strategically masks the initial high-loss tokens, can improve a model's ability to adapt to a new domain while simultaneously mitigating catastrophic forgetting. This method allows the model to focus its updates on the more content-rich portions of the text, where meaningful domain-specific knowledge is more likely to be learned.

## 3.2 THEORETICAL JUSTIFICATION

The core premise of the CA-CPT, is that by systematically removing a known source of noise from the training signal, the stability of the continual learning process can be significantly enhanced. We argue that the initial tokens of a sequence are the primary source of this noise. By treating them as a non-trainable context, we perform a targeted form of gradient variance reduction that links token position to the stability of model parameters.

The optimization dynamics of language models are directly coupled to the information-theoretic properties of sequential data. The conditional entropy of a token's predictive distribution, $H(p_\theta(x_t|x_{<t}))$, quantifies its uncertainty given prior context (Kuhn et al., 2023). For initial tokens ($t = 1$), this context is null, forcing the model to rely on a diffuse, high-entropy unconditional prior. This predictive uncertainty manifests as high-variance gradients through the negative log-likelihood objective, $\mathcal{L}_t(\theta) = -\log p_\theta(x_t|x_{<t})$. The gradient with respect to the logits, $\nabla_{z_t} \mathcal{L}_t = p_t - y_t$, becomes a dense, high-magnitude vector when a flat prediction $p_t$ is contrasted with a one-hot target $y_t$. The variance arises as different initial tokens pull shared parameters in disparate directions, introducing significant noise into the optimization (Chung et al., 2024).

While high gradient variance is a known impediment to convergence in standard training, its effects are amplified in continual learning. In the continual learning setting, the objective is to find parameters that are optimal for a new task B without moving into a high-loss region for a previous task A. The large, stochastic steps induced by high-variance gradients from initial tokens increase the probability of traversing out of a low-loss plateau for prior tasks, directly causing catastrophic forgetting (Wu et al., 2024c).

CA-CPT serves as a principled variance reduction strategy to address this instability. We decompose the total gradient for a sequence, $\nabla \mathcal{L}_{\text{total}}$, into a high-variance component from initial tokens, $\nabla \mathcal{L}_{\text{initial}}$, and a more stable component from subsequent tokens, $\nabla \mathcal{L}_{\text{subsequent}}$. By masking the loss on the initial $k$ tokens, our method effectively nullifies the noisy component $\nabla \mathcal{L}_{\text{initial}}$. The parameter update is thus driven exclusively by the cleaner, more contextually-grounded signal from $\nabla \mathcal{L}_{\text{subsequent}}$. This improves the gradient's signal-to-noise ratio, permitting adaptation to new data while constraining the destructive updates that erode prior knowledge, thereby balancing plasticity and stability.

## 3.3 INTRODUCING CONTEXT-AWARE CONTINUAL PRETRAINING

CA-CPT relies on a strategic preprocessing step: generating and prepending a context to the content of each data sample. This context is then masked during training, focusing the model's learning on the content while providing it with relevant introductory information.

We propose two methods for creating the contexts for CA-CPT. The most suitable approach depends on the dataset's specific characteristics, such as available metadata and document structure.

**Metadata-Based Context Generation.** This method uses existing metadata or structural information to create the context. For documents with titles or abstracts, these elements can serve as the context, as they provide a high-level summary without revealing too much specific information. For non-structural data, we can generate the metadata using a LLM. A key consideration here is to avoid using a detailed summary that could "spoil" the content and inhibit the model's ability to learn from the document itself.

**Empirical Rule-Based Masking.** This method involves masking a fixed portion of the beginning of a document to serve as the context. It's a more generalized approach that doesn't rely on existing metadata. When using this method, it's crucial to balance the amount of text masked with the size of

**No Split**

Maintaining biological integrity requires cells to verify incoming signals before launching a metabolic pathway; if the pathway is inhibited, the system must diagnose the source of the failure. This diagnostic review involves analyzing internal protein markers and genetic expression to determine if the dysfunction was caused by a pathogenic stressor or a catastrophic energy failure leading to apoptosis.

PPL=24.73

**Metadata Split**

Chapter 3: Signal Transduction and Metabolic Integrity
Maintaining biological integrity requires cells to verify incoming signals before launching a metabolic pathway; if the pathway is inhibited, the system must diagnose the source of the failure. This diagnostic review involves analyzing internal protein markers and genetic expression to determine if the dysfunction was caused by a pathogenic stressor or a catastrophic energy failure leading to apoptosis.

PPL=19.94

**Sequential Split**

Maintaining biological integrity requires cells to verify incoming signals before launching a metabolic pathway; if the pathway is inhibited, the system must diagnose the source of the failure. This diagnostic review involves analyzing internal protein markers and genetic expression to determine if the dysfunction was caused by a pathogenic stressor or a catastrophic energy failure leading to apoptosis.

PPL=15.73

**Fixed-ratio Split**

Maintaining biological integrity requires cells to verify incoming signals before launching a metabolic pathway; if the pathway is inhibited, the system must diagnose the source of the failure. This diagnostic review involves analyzing internal protein markers and genetic expression to determine if the dysfunction was caused by a pathogenic stressor or a catastrophic energy failure leading to apoptosis.

PPL=17.73

Figure 2: Context Generation Preprocessing Rules Illustrated. Highlighted words represent the masked context.

the dataset. Masking too large a portion of each sample can lead to a significant loss of information, which can be mitigated by creating multiple versions of the dataset with varying mask ratios.

To create the CA-CPT dataset used in our experiments, we applied a series of specific preprocessing rules to our continual learning datasets. These rules, illustrated in Figure 2 were designed to generate diverse data samples that capture the essence of both context creation methods. Our approach combines these different strategies into a single aggregated dataset to ensure a rich and varied training experience. We used the following specific rules:

- **Metadata Split**: For documents that include them, we used the title and abstract as the context, with the main body of the text serving as the content.
- **Sequential Splits**: We split each document into 10 equal parts. From these, we generated nine new samples. For each new sample, the context consisted of the first $n$ parts, and the content was the $(n+1)$-th part, for $n$ from 1 to 9.
- **Fixed-ratio Splits**: We created three distinct datasets where the context was defined by a fixed percentage of the document's initial tokens:
  - **90/10 Split**: The first 90% of tokens were designated as context, with the final 10% as content.
  - **80/20 Split**: The first 80% of tokens were used as context, with the final 20% as content.
  - **70/30 Split**: The first 70% of tokens were used as context, with the final 30% as content.

Once these CA-CPT datasets are created, they can be used for continual pretraining. During training, we simply ensure that the context portion of any CA-CPT sample is masked, so that the model only computes loss and updates its parameters based on the content part of the sample.

## 4 EXPERIMENTAL SETUP

### 4.1 EXPERIMENTS

We conduct three sets of experiments to evaluate our method. The first set assesses the core performance of CA-CPT on the base Llama 3.1-8B model against a standard CPT approach. This

evaluation quantifies domain learning and catastrophic forgetting by measuring perplexity on new domain-specific datasets and the general-domain RedPajama dataset.

The second set of experiments uses the instruction-tuned Llama 3.1-8B-Instruct model as a baseline to which we added Llama Pro layers, froze all original weights, and trained only the newly added layers. This setup was designed to specifically measure the forgetting of instruction-following skills while the model learns new domain-specific knowledge. It highlights the orthogonality of CA-CPT compared to other catastrophic forgetting mitigation methods.

The third set of experiments relates to downstream tasks. We demonstrate the ability of CA-CPT to maintain performance, i.e. mitigating catastrophic forgetting, on general domain tasks, while also showing performance on question answering tasks on new domains.

Details about our experimental setup can be found in Appendix B respectively.

## 4.2 MODELS

**Llama 3.1-8B.**    We use the base Llama-3.1-8B model, from the Llama 3.1 (Grattafiori et al., 2024) family, to demonstrate the core benefits of CA-CPT. This model serves as the primary control for highlighting how our method mitigates catastrophic forgetting while efficiently acquiring new domain-specific knowledge during continual pre-training.

**Llama 3.1-70B.**    We also use Llama-3.1-70B, which is in the same family, to show that our method works well even when scaling the model size, especially when evaluating on downstream tasks.

**Llama 3.1-8B-Instruct.**    We use an instruction-tuned model which allows us to directly measure how effectively our method mitigates the forgetting of instruction-following abilities while the model learns new domain knowledge.

**Llama 3.1-8B-Instruct + LLaMa Pro (Wu et al., 2024a).**    We combine our method with an architectural method like Llama Pro using Llama 3.1-8B-Instruct to show that CA-CPT can be integrated with other techniques to achieve stronger performance in domain adaptation and catastrophic forgetting mitigation.

## 4.3 DATASETS

To evaluate our method, we selected three distinct and challenging datasets that represent data distributions unlikely to have been encountered by the pretrained model. The goal of this selection was to test the model's ability to adapt to novel linguistic and domain-specific knowledge. We also select one dataset that represents previously acquired knowledge. Specifically, we use:

- **ZelaiHandi** (San Vicente et al., 2024) is the most extensive collection of Basque texts available, ranging from news articles to scientific articles and literature under various CC licenses. The Basque language is particularly well-suited for our experiments due to its unique linguistic properties; it is a unique language with no known genetic relationship to other languages, providing a significant distributional shift for the LLM to adapt to.
- **COLD French Law** (Harvard Library Innovation Lab, 2024) includes over 800,000 French legal articles under CC-BY 4.0 license. It presents a dual specificity, being both in a different language from the model's primary training data and containing highly specialized legal terminology and discourse structures. This combination makes it an excellent test case for cross-lingual and domain-specific adaptation.
- **Climate Policy Radar** (Climate Policy Radar, 2025) contains national law and policy documents submitted to various international environment surveillance organizations such as UNFCCC and NDCs under CC-BY 4.0 license. The specificity of this dataset makes it interesting and a new distribution for the model to deal with.
- **RedPajama** (Together, 2023) is a 30-trillion token dataset released under CC licenses and designed for training LLMs. It encompasses a diverse range of sources and languages and has been a foundational component in the training of many prominent LLMs. In this work, we use this general-domain dataset as a proxy for a model's original pre-training data,

establishing it as our benchmark to measure the catastrophic forgetting of general-purpose knowledge.

Each of the three first datasets was processed and structured according to the CA-CPT methods detailed in Section 3.3.

### 4.4 EVALUATION METRICS

We use perplexity (PPL) (Jelinek et al., 2005) for the CPT experiments to evaluate the quality of our model to predict the next tokens and we use the respective metrics for each dataset to measure the instruction-following capabilities. A lower perplexity score indicates that the model is more confident in its predictions and has a better understanding of the text's underlying structure and vocabulary. We use perplexity to evaluate two key aspects of our approach:

- **Domain Learning:** we calculate the perplexity of our models on the domain-specific datasets (ZelaiHandi, COLD French Law, and Climate Policy Radar). A significant drop in perplexity on these datasets after CPT indicates that the model has successfully learned and adapted to the new domains.

- **Catastrophic Forgetting on General Domain:** we also measure the perplexity on a large, general-domain dataset, the RedPajama dataset. This dataset is representative of the model's original pre-training data. By tracking the perplexity on RedPajama, we directly measure the extent to which our CA-CPT method mitigates the catastrophic forgetting of the model's initial, general-purpose knowledge.

- **Catastrophic Forgetting on Instruction-Following Capabilities:** we use standard benchmarks such as ARC (Clark et al., 2018) under CC-BY-SA 4.0 license, Wino-Grande (Keisuke et al., 2019) under CC-BY 4.0 license, MMLU (Hendrycks et al., 2021) under MIT license, GSM8K (Cobbe et al., 2021) under MIT license, HellaSwag (Zellers et al., 2019) under MIT license, PIQA (Bisk et al., 2020), OpenBookQA (Mihaylov et al., 2018), SciQ (Johannes Welbl, 2017) under CC-BY-NC 3.0 license, and TruthfulQA (Lin et al., 2021) under Apache 2.0 license. The performance on these benchmarks allows us to specifically evaluate how our CA-CPT methodology helps to reduce the forgetting of instruction-following skills while learning new domain-specific knowledge.

- **Downstream Performance on New Domain:** we evaluate the domain adaptation of continually pretrained models on synthetically generated and human annotated downstream tasks. Specifically, we report the accuracy on multiple choices question answering tasks. We expand on the content of these datasets and how they were generated in Appendix C.

## 5 RESULTS

### 5.1 COMPARING CA-CPT TO CPT

Table 1: Catastrophic Forgetting Mitigation on General Domain Data Using Llama 3.1-8B.

| Train Data | Average PPL on RedPajama ($\downarrow$) | | | % Samples where $\text{PPL}_{\text{CPT}} > \text{PPL}_{\text{CA-CPT}}$ |
| --- | --- | --- | --- | --- |
| | Baseline | CPT | CA-CPT | |
| Climate Policy Radar | $12.79_{\pm 1.37}$ | $168.33_{\pm 0.13}$ | $\mathbf{67.37}_{\pm 0.058}$ | 99.83% |
| COLD French Law | $12.79_{\pm 1.37}$ | $93.67_{\pm 0.097}$ | $\mathbf{65.17}_{\pm 0.047}$ | 76.89% |
| Zelai Handi | $12.79_{\pm 1.37}$ | $254.61_{\pm 0.19}$ | $\mathbf{92.60}_{\pm 0.070}$ | 99.53% |

Table 1 shows that on the general-domain RedPajama dataset, CA-CPT demonstrates superior mitigation of catastrophic forgetting. It achieves a significantly smaller increase in perplexity compared to standard CPT. Specifically, the perplexity scores for standard CPT are respectively 2.49×, 1.43×, and 2.74× higher than for CA-CPT when training respectively on the Climate Policy Radar, COLD French Law, and Zelai Handi datasets.

Table 2: Average Perplexity on Domain Adaptation Test Data Using Llama 3.1-8B.

| Train Data | Average PPL on Domain Adaptation Test Data ($\downarrow$) | | |
| --- | --- | --- | --- |
| | Baseline | CPT | CA-CPT |
| Climate Policy Radar | 39.47 $\pm$ 2.44 | **27.77** $\pm$ **0.0059** | 27.79 $\pm$ 0.0059 |
| COLD French Law | 5.18 $\pm$ 0.0034 | **1.43** $\pm$ **0.00021** | 1.71 $\pm$ 0.00026 |
| Zelai Handi | 10.10 $\pm$ 0.038 | **1.63** $\pm$ **0.0052** | 2.74 $\pm$ 0.0016 |

Crucially, this enhanced knowledge retention does not compromise the model's ability to learn new information. As seen in Table 2, on the domain-specific test datasets, both CA-CPT and standard CPT models reduce perplexity to a similar degree, showing that the context-masking strategy does not lead to a significant loss of learning efficiency.

## 5.2 ORTHOGONALITY OF CA-CPT

Table 3: Average Perplexity on Domain Adaptation Test Data Using Llama 3.1-8B-Instruct + Llama Pro.

| Train Data | Average PPL ($\downarrow$) | | |
| --- | --- | --- | --- |
| | Baseline | CPT | CA-CPT |
| Climate Policy Radar | 72.54 $\pm$ 2.03 | **57.14** $\pm$ **1.77** | 59.53 $\pm$ 1.90 |
| COLD French Law | 115.8 $\pm$ 0.42 | 110.84 $\pm$ 0.24 | **96.04** $\pm$ **0.20** |
| Zelai Handi | 137.07 $\pm$ 0.43 | 97.14 $\pm$ 0.35 | **65.06** $\pm$ **0.26** |

Table 3 confirms the dual benefits of CA-CPT: it can effectively be combined with other continual learning methods like Llama Pro. We notice often superior, domain adaptation. For instance, CA-CPT achieves significantly lower perplexity scores on the Climate Policy Radar and Zelai Handi datasets, demonstrating more efficient learning on both of these datasets. This is also shown on downstream tasks in Table 4.

## 5.3 EVALUATION ON DOWNSTREAM TASKS

Table 4: Evaluation on General Knowledge Downstream Tasks Llama 3.1-8B-Instruct + LLaMa Pro.

| Benchmark | Baseline | COLD French Law | | Climate Policy Radar | | Zelai Handi | |
| --- | --- | --- | --- | --- | --- | --- | --- |
| | | CPT | CA-CPT | CPT | CA-CPT | CPT | CA-CPT |
| ARC Challenge | 0.5512 $\pm$ 0.0145 | 0.5094 $\pm$ 0.0146 | **0.5162** $\pm$ 0.0146 | 0.5171 $\pm$ 0.0146 | **0.5461** $\pm$ 0.0145 | 0.5077 $\pm$ 0.0146 | **0.5128** $\pm$ 0.0146 |
| ARC Easy | 0.7984 $\pm$ 0.0083 | 0.7538 $\pm$ 0.0088 | **0.7626** $\pm$ 0.0087 | 0.7155 $\pm$ 0.0093 | **0.7934** $\pm$ 0.0083 | 0.7319 $\pm$ 0.0091 | **0.7437** $\pm$ 0.0090 |
| Hellaswag | 0.7925 $\pm$ 0.0041 | **0.7789** $\pm$ 0.0041 | 0.7739 $\pm$ 0.0042 | 0.7867 $\pm$ 0.0041 | **0.7876** $\pm$ 0.0041 | 0.7639 $\pm$ 0.0042 | **0.7657** $\pm$ 0.0042 |
| OpenBookQA | 0.4300 $\pm$ 0.0222 | **0.4180** $\pm$ 0.0221 | 0.4140 $\pm$ 0.0220 | **0.4360** $\pm$ 0.0222 | 0.4340 $\pm$ 0.0222 | 0.3860 $\pm$ 0.0218 | **0.4220** $\pm$ 0.0221 |
| PIQA | 0.8085 $\pm$ 0.0092 | 0.7992 $\pm$ 0.0093 | **0.8020** $\pm$ 0.0093 | 0.7938 $\pm$ 0.0094 | **0.8036** $\pm$ 0.0093 | 0.7753 $\pm$ 0.0097 | **0.7856** $\pm$ 0.0096 |
| SciQ | 0.9610 $\pm$ 0.0061 | 0.9450 $\pm$ 0.0072 | **0.9490** $\pm$ 0.0070 | 0.9310 $\pm$ 0.0080 | **0.9590** $\pm$ 0.0063 | **0.9480** $\pm$ 0.0070 | 0.9470 $\pm$ 0.0071 |
| TruthfulQA MC2 | 0.5413 $\pm$ 0.0150 | 0.5160 $\pm$ 0.0153 | **0.5217** $\pm$ 0.0153 | **0.5546** $\pm$ 0.0152 | 0.5453 $\pm$ 0.0151 | **0.5325** $\pm$ 0.0153 | 0.5241 $\pm$ 0.0152 |
| WinoGrande | 0.7356 $\pm$ 0.0124 | **0.7395** $\pm$ 0.0123 | 0.7293 $\pm$ 0.0125 | 0.7088 $\pm$ 0.0128 | **0.7419** $\pm$ 0.0123 | 0.7009 $\pm$ 0.0129 | **0.7238** $\pm$ 0.0126 |
| GSM8K | 0.7809 $\pm$ 0.0117 | 0.6846 $\pm$ 0.0128 | **0.7043** $\pm$ 0.0126 | 0.6975 $\pm$ 0.0127 | **0.7521** $\pm$ 0.0119 | 0.5406 $\pm$ 0.0137 | **0.6520** $\pm$ 0.0131 |
| MMLU | 0.6818 $\pm$ 0.0037 | 0.6763 $\pm$ 0.0038 | **0.6782** $\pm$ 0.0038 | 0.6743 $\pm$ 0.0038 | **0.6815** $\pm$ 0.0037 | 0.6534 $\pm$ 0.0038 | **0.6691** $\pm$ 0.0038 |

As we can see in Table 4, CA-CPT generally outperforms standard CPT on downstream tasks on general domain. This means that, in addition to having lower perplexity on our general knowledge dataset, models trained with CA-CPT can be expected to perform better on previously learned tasks.

Finally, Table 5 highlights the trade-off introduced by CA-CPT between retaining previously learned knowledge and adapting to new downstream tasks. We can see that applying CA-CPT effectively allows our model to perform well on downstream tasks in all kinds of settings. For example, on

Table 5: Evaluation on Domain Specific Downstream Multiple Choices Question Answering Tasks.

| Model | Task | Accuracy (↑) | | |
|---|---|---|---|---|
| | | Baseline | CPT | CA-CPT |
| Llama 3.1-8B | Climate Policy Radar | $0.2561 _{\pm 0.0485}$ | $\textbf{0.5000} _{\pm \textbf{0.0556}}$ | $0.4024 _{\pm 0.0545}$ |
| | COLD French Law | $0.3095 _{\pm 0.0413}$ | $\textbf{0.6270} _{\pm \textbf{0.0433}}$ | $0.6032 _{\pm 0.0438}$ |
| Llama 3.1-8B-Instruct | Climate Policy Radar | $0.2561 _{\pm 0.0485}$ | $\textbf{0.5854} _{\pm \textbf{0.0547}}$ | $0.3780 _{\pm 0.0539}$ |
| | COLD French Law | $0.3175 _{\pm 0.0416}$ | $0.5952 _{\pm 0.0439}$ | $\textbf{0.6349} _{\pm \textbf{0.0431}}$ |
| Llama 3.1-8B-Instruct + LLaMa Pro | Climate Policy Radar | $0.2561 _{\pm 0.0485}$ | $\textbf{0.4634} _{\pm \textbf{0.0554}}$ | $0.3902 _{\pm 0.0542}$ |
| | COLD French Law | $0.3175 _{\pm 0.0416}$ | $\textbf{0.5238} _{\pm \textbf{0.0447}}$ | $0.4444 _{\pm 0.0444}$ |
| Llama 3.1-70B | Climate Policy Radar | $0.3537 _{\pm 0.0531}$ | $\textbf{0.6829} _{\pm \textbf{0.0517}}$ | $0.5000 _{\pm 0.0556}$ |
| | COLD French Law | $0.3730 _{\pm 0.0433}$ | $0.6746 _{\pm 0.0419}$ | $\textbf{0.6905} _{\pm \textbf{0.0413}}$ |

COLD French Law, both Llama 3.1-8B-Instruct and Llama 3.1-70B trained with CA-CPT outperform standard CPT.

## 6    LIMITATIONS

The effectiveness of CA-CPT is dependent on the dataset's structure. For unstructured data, applying our metadata-based context creation method can become computationally intensive at scale, as it would require us to synthetically generate the inexistent metadata. The alternative, empirical rule-based masking, offers more flexibility but requires careful tuning to be effective. Moreover, our experimental results are currently confined to the Llama 3.1 model family. While the findings are strong, further research is required to verify that our method generalizes effectively across a wider range of model families.

## 7    CONCLUSION

In this work, we introduced Context-Aware Continual Pretraining, a simple, powerful, and generalizable method to mitigate catastrophic forgetting when continually pretraining LLMs. The core insight is that the initial tokens contribute disproportionately to gradient variance, destabilizing the learning process and leading to the erasure of prior knowledge. By strategically masking these tokens from the loss computation, CA-CPT provides a more stable training signal, allowing the model to effectively acquire new information without catastrophically forgetting its original knowledge. Through empirical validation, we proved that CA-CPT significantly improves the stability-plasticity trade-off compared to standard baselines. Indeed, our approach consistently improved the retention of both general knowledge and instruction-following capabilities, while achieving on-par or superior performance when adapting to new domains. CA-CPT represents a valuable and practical contribution to the ongoing effort to build more robust, adaptable, and truly lifelong learning artificial intelligence systems.

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

## A  USE OF LARGE LANGUAGE MODELS IN PAPER WRITING

We disclose the use of LLMs to polish writing. Mostly, we used LLMs to make sentences more concise and readable and to generate LaTeX formatted tables.

## B  TRAINING SETUP AND HYPERPARAMETERS

All experiments were conducted on NVIDIA A100 GPUs and with identical hyperparameters to ensure a fair comparison. We use a customized version of the LLaMA-Factory (Zheng et al., 2024) framework, under Apache 2.0 license, adapted for CA-CPT.

Table 6: Detailed Experimental Setup for Each Training Run.

| Training Parameter | Value |
|---|---|
| batch size | 64 |
| training epochs | 1 |
| learning rate | $2 \times 10^{-5}$ |
| warmup ratio | 0.1 |
| learning rate schedule | cosine |
| optimizer | AdamW |
| *for the Llama Pro experiments* | |
| number of llama pro layers | 8 |
| llama pro layers positions | $[18, 21, 24, 27, 30, 33, 36, 39]$ |

## C  GENERATION OF SYNTHETIC DATASETS FOR DOWNSTREAM TASKS

To generate datasets for downstream domain tasks, we sample documents from the domain training set. Using Llama 3.1-70B-Instruct, we generate one question per sampled document, For each question, we also generate one true answer based on the text in the document and three false answers. Prompts to generate the questions and the answers are presented in Figures 3 to 5.

We generate 300 questions and associated answers for each dataset. Then, we manually review the questions and answers, verifying the format and the truthfulness of the answers. We also filter out samples that with trivial or unsatisfying questions and answers. In total, our datasets contain 126 questions for COLD French Law and 82 questions for Climate Policy Radar. We have not created a test dataset for ZelaiHandi since none of the authors understand the Basque language.

```
You are an expert content creator specializing in generating
multiple-choice test questions. Your task is to analyze a
given text and compose a single, specific factual question
based on the information provided. The question must be
well-grounded and non-trivial, meaning it should require
the reader to understand a definition, a relationship, a
responsibility, or a process described in the text, rather
than simply recalling a single number, date, or name.

**Instructions:**
- **Role:** You are acting as a question generator.
Do not provide an answer.
- **Output:** Your output must be **only** the question
itself.Do not include any preambles, introductory
phrases, or explanations.
- **Clarity:** The question must be clear, concise,
and directly solvable using only the information in the
provided text.
- **Focus:** The question should test a key concept,
definition, or relationship within the text.
- **Format:** The question must end with a question mark (?).

**Text:**
{text}

**Question:**
```

Figure 3: Prompt to Generate a Question from a Document

```
You are an expert fact-checker and information extractor. Your
sole purpose is to provide the correct, factual answer to a
given question based **only** on the information within the
provided text.

**Instructions:**
- **Role:** You are acting as a precise, automated information extractor.
- **Output:** Your output must contain **only** the factual answer.
Do not include any preambles (e.g., "The answer is..."), conversational
filler, or explanations.
- **Accuracy:** The answer must be a direct and truthful response based
**solely** on the provided text.
- **Conciseness:** Provide the answer in a single, short sentence unless
the information is a number.

**Text:**
{text}

**Question:**
{question}

**Answer:**
```

Figure 4: Prompt to Generate a True Answer from a Document and a Question

```
You are an expert at creating misleading but plausible incorrect answers
for multiple-choice questions. Your task is to generate a single,
factually incorrect answer based on the provided text and question. The
incorrect answer must be able to fool a human into believing it's correct.

**Instructions:**
- **Role:** You are a misinformation generator. Your only output should
be a plausible, but false, answer.
- **Output:** Your output must be **only** the single, incorrect answer.
Do not include any preambles (e.g., "The answer is..."), conversational
filler, or explanations.
- **Plausibility:** The false answer should appear convincing. It should
not be an obvious falsehood.
- **Conciseness:** Provide the answer in a single, short sentence unless
the information is a number.
- **Uniqueness:** The incorrect answer must be distinct from the following
list of previous answers: {previous_answers}

**Text:**
{text}

**Question:**
{question}

**Correct Answer:**
{question}

**Incorrect Answer:**
```

Figure 5: Prompt to Generate a False Answer from a Document and a Question

