# OpenReview forum: "Mitigating Catastrophic Forgetting with Context-aware Continual Pretraining for LLMs"
_ICLR.cc/2026/Conference — ICLR 2026 Conference Withdrawn Submission_

### Official Review · Reviewer_YZYg · 2025-10-30

**Soundness:** 2
**Presentation:** 2
**Contribution:** 2
**Rating:** 2
**Confidence:** 3

**Summary:**

This paper proposes Context-Aware Continual Pretraining (CA-CPT), a simple yet effective method that mitigates catastrophic forgetting in large language models by masking high-loss initial tokens during continual pretraining. The method achieves improved stability and adaptation on new domains while maintaining general and instruction-following capabilities.

**Strengths:**

1. The paper provides clear theoretical reasoning for why initial tokens incur high loss and logically explains how the proposed method addresses this issue.
2. CA-CPT can be easily integrated into existing continual pre-training (CPT) methods, demonstrating strong scalability.

**Weaknesses:**

1. As mentioned in the limitation section, this work exhibits critical weaknesses, including limited experimental diversity with evaluations confined to the LLaMA 3.1 family, and reliance on empirical rule-based masking that introduces strong tuning dependencies.
2. The paper does not include lifelong continual pretraining experiments or forward/backward transfer metrics as established in prior CPT benchmarks (e.g., Jin et al., 2022).
3. Comparative studies with other continual pretraining paradigms, such as replay-based and regularization-based methods, are insufficient.

**Questions:**

1. Is instability truly caused by the high gradient variance of initial tokens, and if so, have alternative approaches such as gradient scaling rather than masking been explored?
2. Does the gradient stability and learning efficiency remain consistent when CA-CPT is combined with PEFT methods such as LoRA or Adapter tuning?

---

### Official Review · Reviewer_jysg · 2025-10-31

**Soundness:** 3
**Presentation:** 3
**Contribution:** 2
**Rating:** 4
**Confidence:** 4

**Summary:**

This paper aims to address the catastrophic forgetting problem encountered during continuous pretraining in LLM (Laminated Learning Model). The authors observed that the initial few tokens of a sequence generate disproportionately high losses during training and inferred that the resulting gradient variance is a key factor destabilizing the model. Based on this, the paper proposes a method called Context-aware Continual Pretraining (CA-CPT). This method defines the initial portion of each sample as the context during data preprocessing and masks it in loss calculation, thus allowing model updates to be driven only by subsequent content. The authors validated the method through a series of experiments on the Llama 3.1 model family. Experimental results show that, compared to standard CPT, CA-CPT exhibits a lower forgetting rate on general knowledge datasets and standard benchmarks, while achieving comparable or slightly worse performance in acquiring knowledge from new domains.

**Strengths:**

1. Clear motivation. The core observation of the paper—the high loss of the initial tokens of the sequence (as shown in Figure 1)—is clear and intuitive. Connecting this observation to gradient variance and training instability provides a reasonable (albeit heuristic) theoretical foundation for the proposed method.

2. Simplicity and practicality. The proposed CA-CPT method is very simple, easy to understand and implement. It is essentially a data preprocessing technique that does not require modification of the model architecture or complex training procedures.

4. Potential in mitigating forgetting. Experimental results show that CA-CPT consistently outperforms the standard CPT baseline in preserving general knowledge and instruction compliance. This demonstrates the effectiveness of the method in improving model "stability".

**Weaknesses:**

1. The authors observed a high loss for the initial tokens of a sequence during model pre-training. The model's loss for each sample varies across different training stages and tends to converge. I'm unclear on which training stage the reported sample represents, and whether the authors could provide the losses for different samples at different training stages to better illustrate their observations.

2. In the theoretical explanation, the argument directly links high gradient variance to catastrophic forgetting, which is logically weak. It doesn't **formally** explain why this instability "directly leads" to catastrophic forgetting. There are multiple causes of forgetting and multiple methods to address it; the author's stated goal may only be one explanation and cannot be generalized. Connecting the observed phenomenon to the essence of catastrophic forgetting requires more mathematical proof than textual explanation.

3. The authors proposed several different segmentation methods, but these are mostly simple, experience-based, or predefined proportions. Are there more automated methods for better segmentation?

4. The paper only reports the absolute performance of the proposed method but does not provide a systematic comparison with current mainstream catastrophic forgetting mitigation methods. Therefore, it is difficult for readers to determine the actual competitiveness and advantages of this method within the existing technology. Table 1 shows that its effect on mitigating catastrophic forgetting is limited.

5. In Table 1, the authors need more general benchmarks to demonstrate the mitigation of catastrophic forgetting, rather than just the RedPajama dataset; for example, MMLU-Pro or GSM8k could be used.

6. The authors want to demonstrate that CA-CPT can be used simultaneously with other methods for mitigating catastrophic forgetting, so more other methods are needed for verification, not just one.

7. If the authors' observations are valid, I am curious whether this method remains effective in SFT or reinforcement learning stages? Why does this paper only work on CPT?

**Questions:**

see weakness

---

### Official Review · Reviewer_a2fb · 2025-11-01

**Soundness:** 2
**Presentation:** 3
**Contribution:** 2
**Rating:** 4
**Confidence:** 4

**Summary:**

CA-CPT is a flexible technique that provides the model with sample-specific context before adapting its weights to new content, helping to smooth the training loss.

**Strengths:**

1. This work illustrates that different splits bring different perplexities.

2. This work is easy to follow for the authors' presentation.

3. The different scales of LLM models are evaluated to prove the generalizability.

**Weaknesses:**

1. The rationale between the impact of context and continual learning is not explicitly quantified.

2. The independently proposed insights are insufficient.  The noise in the starting tokens enhances gradient variance reduction that links token position to the stability of model parameters. However, this is mainly proposed by [1].

3. The authors claim the proposed CA-CPT is orthogonal to existing catastrophic forgetting mitigation strategies. However, no empirical evaluations are provided.

4. The baseline methods are limited. The existing LLM methods that mitigate catastrophic forgetting should be considered.

5. CA-CPT cannot achieve consistent improvements in terms of the empirical results.


[1] Chung, Woojin, et al. "Stable Language Model Pre-training by Reducing Embedding Variability." arXiv preprint arXiv:2409.07787 (2024).

**Questions:**

See weakness above.

---

### Official Review · Reviewer_2X8i · 2025-11-02

**Soundness:** 2
**Presentation:** 3
**Contribution:** 3
**Rating:** 4
**Confidence:** 3

**Summary:**

This paper studies the catastrophic forgetting issues during the continue pre-training of LLMs. In particular, this paper proposes Context-Aware Continual Pretraining (CA-CPT), a method which masks high-loss initial tokens during continual pretraining, CA-CPT stabilizes learning, retains prior knowledge, and efficiently adapts models to new domains, outperforming standard approaches without sacrificing new-domain performance.

**Strengths:**

- The proposed CA-CPT is conceptually simple, easy to implement, and incurs minimal additional computational cost compared with other continual learning strategies.
- The paper presents a well-motivated theoretical and empirical analysis, clearly explaining the rationale behind masking high-loss initial tokens and linking it to gradient variance reduction and stability–plasticity trade-off.
- The method proposed in this paper is orthogonal and complementary to existing CPT approaches (e.g., replay-based, regularization, and architectural methods), allowing flexible integration for further performance gains.

**Weaknesses:**

- The experiments are limited to the LLaMA model family, which weakens generality of the conclusions. The motivation and empirical evidence for CA-CPT’s effectiveness would be stronger if validated across diverse architectures.
- The data preprocessing procedure lacks transparency. Although three data generation strategies (metadata split, sequential split, and fixed-ratio split) are mentioned, the paper does not clearly explain the mixing logic or how these datasets were combined for training.
- The paper does not analyze the sensitivity of CA-CPT to context construction methods or hyperparameters, such as the length and ratio of masked tokens, which may critically influence model learning and forgetting behaviors. This omission reduces the practical reproducibility and applicability of the approach.
- The orthogonality analysis only considers integration with LLaMA Pro, leaving open questions about the method’s compatibility and generalizability to other continual learning frameworks.
- The discussion of downstream performance is selective and somewhat misleading. As shown in Table 5, CA-CPT often underperforms CPT on several tasks, yet the paper highlights only a few favorable cases (e.g., COLD French Law) to claim broad effectiveness. This raises concerns about result interpretation bias and the method’s robustness.

**Questions:**

N/A

---

### Note · Authors · 2025-12-03

**Comment:**

We thank the reviewers for their thorough evaluation. After internal discussion, we have decided to withdraw this submission. Your constructive feedback will be invaluable in refining the manuscript for the future.

**Withdrawal Confirmation:**

I have read and agree with the venue's withdrawal policy on behalf of myself and my co-authors.